# Effect of Combined High-Pressure Homogenization and Biotechnological Processes on Chitin, Protein, and Antioxidant Activity of Cricket Powder-Based Ingredients

**DOI:** 10.3390/foods13030449

**Published:** 2024-01-31

**Authors:** Samantha Rossi, Davide Gottardi, Alberto Barbiroli, Mattia Di Nunzio, Lorenzo Siroli, Giacomo Braschi, Oliver Schlüter, Francesca Patrignani, Rosalba Lanciotti

**Affiliations:** 1Department of Agricultural and Food Sciences, Alma Mater Studiorum, University of Bologna, Piazza Goidanich 60, 47521 Cesena, Italylorenzo.siroli2@unibo.it (L.S.); giacomo.braschi2@unibo.it (G.B.); oschlueter@atb-potsdam.de (O.S.); francesca.patrignani@unibo.it (F.P.); rosalba.lanciotti@unibo.it (R.L.); 2Interdepartmental Centre for Agri-Food Industrial Research, University of Bologna, 47521 Cesena, Italy; 3Department of Food, Environmental and Nutritional Sciences (DeFENS), University of Milan, Via Celoria 2, 20133 Milan, Italy; alberto.barbiroli@unimi.it (A.B.); mattia.dinunzio@unimi.it (M.D.N.); 4Leibniz Institute for Agricultural Engineering and Bioeconomy, Quality and Safety of Food and Feed, Max-Eyth-Allee 100, 14469 Potsdam, Germany

**Keywords:** cricket powder, *Debaryomyces hansenii*, *Yarrowia lipolytica*, chitin content, antioxidant activity, accessible thiols, protein

## Abstract

The main objective of this work was to evaluate the combined effect of a biotechnology process, based on selected yeast strains, and a high-pressure homogenization (HPH) treatment on the microbiological quality, structural organization of proteins, chitin content, and antioxidant activity of a mixture of cricket powder (*Acheta domesticus*) and water. Compared to untreated samples, the cricket matrix treated with HPH four times at 180 MPa promoted the growth of the inoculated *Yarrowia lipolytica* and *Debaryomyces hansenii* strains. HPH did not affect the concentration of chitin; however, the combination with microorganisms tended to reduce the content. Although the antioxidant activity increased from 0.52 to 0.68 TAC mM/TE after a 48 h incubation in the control, it was further improved by the combination of HPH and *D. hansenii* metabolism, reaching a value of 0.77 TAC mM/TE. The combination of the two approaches also promoted a reduction in the intensity of bands with molecular weights between 31 and 21.5 kDa in favor of bands with a lower molecular weight. In addition, HPH treatment reduced the number of accessible thiols, suggesting protein structure changes that may further impact the technological properties of cricket powder.

## 1. Introduction

In the perspective of a world population increase, insects are a promising alternative to traditional livestock for addressing the expected animal protein-based or alternative protein-based product demand. Edible insects are usually characterized by a similar or higher protein, fat, mineral, vitamin, and energy content than conventional foods of animal origin. In particular, they contain more polyunsaturated fatty acids and have a higher amount of minerals such as iron and zinc [1]. In addition, edible insect breeding is more sustainable than conventional breeding because it requires a very small area of land, produces lower greenhouse gas and ammonia emissions, and consumes less water and food [2]. However, despite the beneficial aspects related to the consumption and breeding of insects, the idea of introducing them into Western countries’ diet is not well perceived by consumers. The use of insect powder or flour to develop familiar foods such as bread, pasta, biscuits, crackers, burgers, etc., could be a good strategy to reduce the aversion to entomophagy and increase consumer acceptance [3,4]. In addition, to further increase the use of this alternative source, it is essential to design and optimize specific technological and biotechnological processes and formulations to ensure food safety of insect products and encourage scaling-up of insect production at an industrial level. The literature suggests that the production of cricket hydrolysates using safe microorganisms allows the production of sustainable food ingredients with improved sensory and functional properties. In particular, the use of *Yarrowia lipolytica* and *Debaryomyces hansenii* in mixtures of cricket powder and water increase the food safety, functionality, and sensory and technological properties of these mixtures [5]. Specifically, *Y. lipolytica* RO25 hydrolyzed cricket-based ingredients for producing sourdough for bakery goods are characterized by a specific sensory and qualitative characteristic as well as proteolytic, fatty acid, and volatile molecule profiles and the lowest level of biogenic amines when compared to a sample obtained without microbial hydrolysis [6,7]. Other authors have reported that the use of some non-thermal technologies may be a suitable pretreatment for the valorization of *Acheta domesticus* biomass with possible industrial application. Psarianos et al. [8] showed the ability of pulsed electric fields to increase the extraction yields of proteins and fats as well as the oil binding and emulsifying capacity and the antioxidant activity of house cricket powder. Instead, Ugur et al. [9] used high hydrostatic pressures to extract oils from *Acheta domesticus*, demonstrating the desirable physicochemical characteristics of the obtained oils useful as food ingredients. Among non-thermal technologies, high-pressure homogenization (HPH) is a widely used technique in the pharma industry and is applied at a low level of pressure in the food industry. The main use of the HPH technique in the dairy, pharmaceutical, and cosmetic industries, is aimed at reducing particle sizes and increasing the stability of emulsions. However, HPH can reduce microbial cell loads, preserving the physicochemical, nutritional, and sensory properties of the raw materials and ingredients used [10]. Another important aspect of HPH treatment is the ability, in relation to the level of pressure applied, to modify the functional and active properties of food constituents, particularly proteins and enzymes [11,12]. Moreover, the ability of HPH to improve the microstructure of food, rheology, and availability of bioactive compounds [10,13,14] makes this an interesting technique to apply to an innovative matrix such as cricket powder. There are no literature data available regarding the effects of HPH on cricket powder and on the effect of microorganisms’ inoculation in this matrix upon HPH treatment. However, HPH can impact the structure of proteins, making them more bioavailable or reducing their potential toxic effect. For instance, Panozzo et al. [15] reported a decrease in immunoreactivity of egg white due to changes in protein structure upon HPH treatment at 150 MPa for multiple passes. Eventually, HPH can effectively reduce the size of natural fibers, producing nanofibers from cellulose and chitin [16,17]. The cellulose or chitin suspensions are usually subjected to multiple passes (20–30) through the homogenizer at pressures ranging from 100 to 150 MPa [17], and the nanofiber sizes obtained are dependent on the number of passes and pressure [18]. Since cricket powder contains chitin, the impact of HPH on this constituent is also very important. In fact, HPH treatment (60–80 MPa) can reduce the crystallinity and improve enzymatic hydrolysis efficiency of chitin increasing the concentration of N-Acetyl-glucosamine [19,20].

For all these reasons, the objective of the present study was to assess the combined effect of HPH treatment (performed at 180 MPa) and biotechnological process, carried out by *Y. lipolytica* and *D. hansenii* strains previously selected and tested by Patrignani et al. [5] and Rossi et al. [6], on a mixture of cricket powder (*Acheta domesticus*) and water in order to evaluate any changes in microbiology quality and pH, protein structural organization, chitin content, and antioxidant activity. The results obtained in this work showed that the combination of the two approaches promoted the production of innovative ingredients with high functional and technological properties.

## 2. Materials and Methods

### 2.1. Growth Conditions of Yeast Strains

*Yarrowia lipolytica* RO25, *Yarrowia lipolytica* PO11, and *Debaryomyces hansenii* SP6L12, belonging to the strain collection of the Department of Agricultural and Food Sciences, University of Bologna, were used to hydrolyze the cricket powder as reported by Patrignani et al. [5]. Before their use, *Yarrowia* and *Debaryomyces* strains were grown twice separately in YPD (Thermo Fisher, Basigstone, UK) broth and incubated at 25 °C for 48 h with agitation.

### 2.2. Preparation of the Cricket Powder Hydrolysate and HPH Treatment

According to Patrignani et al. [5], cricket powder was mixed with water at a ratio of 1:3 (*w*/*v*). Half of the obtained mixture was treated with HPH, the remaining part was used as the not-treated control. Homogenizing treatment was performed using a PANDA continuous high-pressure homogenizer (GEA, Parma, Italy).

Before high-pressure treatment, the matrix was passed through the homogenizer at 0.1 MPa to favor a greater homogenization of the product. The high-pressure treatment was performed four times at 180 MPa. The time of treatment was constant and instantaneous (1.6 ms). The pressure applied was selected based on Huang et al. [21], who showed that 180 MPa treatment, and not lower pressures, applied for three passes disrupted completely the filamentous mycelium, which is known to contain a not negligible amount of chitin [22], and increased the protein concentration. Due to the complexity of the cricket powder with respect to the fungal matrix, four passes were selected. The treated mixture was cooled by using a thermal exchanger (GEA, Parma, Italy). 

The treated (-HPH) and not-treated (-NT) cricket powder mixtures were divided into six sterile flasks and were inoculated separately with *Y. lipolytica* RO25 (RO25-HPH and RO25-NT), *Y. lipolytica* PO11 (PO11-HPH and PO11-NT), and *D. hansenii* SP6L12 (Db-HPH and Db-NT) at a level of 5/6 Log10 CFU/g. The treated (NoH-HPH) and not-treated (NoH-NT) control samples were prepared in the same way but without yeast inoculation.

The eight samples were incubated at 25 °C for 48 h with agitation (150 rpm).

### 2.3. Microbiological Analysis and pH

Microbiological analyses were performed on the HPH-treated and not-treated cricket powder mixtures immediately after inoculation and after 24 and 48 h of incubation. 

The cellular load of yeasts used in the trial was monitored on yeast extract peptone dextrose (YPD) agar with 0.02% chloramphenicol. Serial decimal dilutions were performed on 1 mL of sample and subsequently plated on YPD. Inoculated plates were incubated at 25 °C for 48 h.

The pH values were detected during the incubation of the samples with a pH meter (SevenCompact S220 pH meter; Mettler Toledo, Urdorf, Switzerland).

### 2.4. Determination of Chitin Content

To assess the effect of the biotechnological approach and the HPH treatment on the chitin contained in the samples, the chitin content was evaluated according to Zamani et al. [23]. Samples were freeze-dried in an Alpha 2–4 LD freeze-dryer (Christ, Osterode am Harz, Germany), and chitin detection was performed directly on the lyophilized powder.

HPH-treated and untreated not-fermented samples (NoH-HPH and NoH-NT) were used as control samples. The impact of yeast strains on the chitin content was evaluated on HPH-treated and not-treated samples obtained after 48 h of incubation. Chitin from shrimp shells (C7170, Merck) was used for the calibration curve, and the results were expressed as mg chitin/100 mg samples based on dry weight.

### 2.5. Protein Characterization

The protein pattern of the samples was characterized by sodium dodecyl sulphate–polyacrylamide gel electrophoresis (SDS-PAGE). Electrophoresis was carried out on polyacrylamide gels following the method described by Laemmli [24].

Samples were prepared by adding 100 µL of sodium phosphate buffer 50 mM, containing 0.1 M NaCl, SDS 2%, and DTT 10 mm at 5 mg of each freeze-dried sample. After 10 min of heat treatment at 100 °C, 100 µL of denaturing buffer (0.125 M Tris-HCl, pH 6.8, 50% (*w*/*v*) glycerol, 7 g/L SDS, 0.1 g/L bromophenol blue, and 10 g/L 2-mercaptoethanol) was added and heated again at 100 °C for 10 min. The denatured samples were centrifuged at 10,000× *g* for 3 min. The electrophoretic separation was performed by running 12 μL of the clear solutions onto a 12% polyacrylamide gel in a Miniprotean II cell (Bio-Rad Laboratories, Hercules, CA, USA), with running buffer 0.025 M Tris-HCl, 0.192 M glycine, and 1 g/L SDS. The gels were Coomassie Blue-stained.

### 2.6. Total Accessible Thiols

The quantitative determination of thiols was carried out using the reagent 2,2′-dinitro-5,5′-dithiodibenzoic acid (DTNB) [25].

Accessible thiols were measured according to Barbiroli et al. [26] with some modifications. 

Forty milligrams of freeze-dried samples were suspended in 5 mL of 50 mM sodium phosphate buffer, pH 7, containing 0.1 M NaCl and 0.2 Mm DTNB, with or without 2% SDS. After 20 and 60 min of incubation with agitation and in the dark, insoluble material was removed by centrifugation at 13,000× *g*, 15 °C, for 15 min. The absorbance of the supernatant was read at 412 nm against a DTNB blank. 

### 2.7. In Vitro Total Antioxidant Capacity (TAC) Determination 

TAC was assessed by the ABTS and DPPH assays on the basis of the ability of the antioxidant molecules to reduce the radical cation of 2,20-azino-bis-(3-ethylbenzothiazoline-6-sulfonic acid) (ABTS) and 2,2-diphenyl-1-picrylhydrazyl (DPPH) [27]. Ten microliters of sample extract were added to 990 µL of 80 µM ABTS+ and 100 µM·DPPH, and the quenching of the absorbance at 734 nm for 1 min and at 517 nm for 30 min for ABTS·+ and DPPH·, respectively, was monitored. Values obtained for each sample were compared to the concentration–response curve of the standard Trolox solution and expressed as mM of Trolox equivalent (TE). 

### 2.8. Statistical Analysis

The results are expressed as the mean ± standard deviation of three independent replicates. The raw data were statistically compared using the one-way ANOVA with a level of significance of 0.05. The differences between mean values were detected with the HSD Tukey test. The software used for the analyses was IBM SPSS Statistics 23 (IBM Corp., Armonk, NY, USA).

## 3. Results and Discussion

### 3.1. Microbiological Quality and pH

Microbiological analyses allowed an evaluation of the growth of the yeasts inoculated during 48 h of incubation. As shown in Figure 1, the HPH treatment applied (180 MPa × four cycles) reduced the initial indigenous yeast load of the starting matrix while promoting the growth of the inoculated yeasts. HPH-treated not-inoculated samples (NoH-HPH, control) had a yeast cell load under the detection limit for all the 48 h of incubation at 25 °C. This is supported by several studies that demonstrated the inactivation of yeast cells when HPH treatments were applied at a lower level of pressure [28]. HPH treatment positively affected the growth of the inoculated yeasts. In fact, HPH-treated and inoculated samples showed higher microbial cell loads than not-treated inoculated samples regardless of incubation time. In particular, samples inoculated with *D. hansenii* showed the highest microbial growth rates when treated with HPH than samples inoculated with *Y. lipolytica* strains. Db-HPH reached a final cell load of 8.6 log CFU/g after 48 h of incubation compared to Db-NT, which reached values of 7.3 log CFU/g. HPH-treated samples obtained from *Y. lipolytica* PO11 and RO25 reached 7.9 and 7.5 log CFU/g after 48 h, respectively, while not-treated samples reached values of 7.2 and 6.9 log CFU/g. The increase in microbial growth reported in Figure 1 may depend on the HPH treatment, which is able to increase the bioaccessibility of nutrients contained in the starting matrix increasing their bioavailability towards the metabolism of inoculated yeasts. Indeed, several studies reported the ability of HPH to increase the nutritional quality of a wide range of foods [29,30]. For example, Di Nunzio et al. [31] reported that HPH improves the nutritional quality of plant food, while the effect on antioxidant and vitamin availability depended on the food matrix. Benjamin and Gamrasni [32] also reported that pomegranate juice treated with HPH had a higher antioxidant value and TPC levels than untreated juice. The pH data reported lower values in HPH-treated samples than in untreated samples. In fact, after 48 h, the pH values of RO25-HPH, PO11-HPH, and Db-HPH were significantly lower (5.55 ± 0.02, 5.76 ± 0.01, and 5.74 ± 0.03, respectively) than those of RO25-NT, PO11-NT, and Db-NT (5.75 ± 0.01, 5.92 ± 0.03, and 5.99 ± 0.01, respectively). The range of pH reached upon incubation was in line with what was reported by Patrignani et al. [5]. The same authors suggested that this reduction could be achieved through deacetylation of chitin performed by microorganisms. 

### 3.2. Chitin Content

The predominant form of chitin in nature, found in the exoskeleton of insects, is α-chitin. The α-form is highly crystallized and more difficult to degrade than β-chitin in enzymatic hydrolysis. To reduce the crystalline structure of chitin and promote its hydrolysis, HPH treatment could be considered a valid technique. Homogenization of chitin suspensions to produce chitin nanofibers, has been widely studied and reviewed [17,33,34,35]. 

The chitin content (expressed as mg/100 mg dry weight) of HPH-treated and not-treated samples obtained after 48 h from the inoculum of *Y. lipolytica* RO25, *Y. lipolytica* PO11, and *D. hansenii* SP6L12 was compared with the not-inoculated samples immediately after (0 h) and after 48 h of incubation (Figure 2). When compared with the not-hydrolyzed mixtures (NoH- samples), the results showed no significant differences among samples regardless of the treatment and the strain applied. In fact, the concentration of chitin in the NoH-NT (0 h) sample was 10.56 mg chitin /100 mg d.w., similar to the one measured for the NoH-HPH (0 h) sample (10.87 mg chitin /100 mg d.w). However, according to the literature, HPH should be highly effective at reducing the crystallinity of chitin, making it more exposed to hydrolysis. In fact, Wei et al. [20] and Zhai et al. [19] showed that HPH treatment applied on crayfish shell at 40 MPa for five steps or on chitin at 60 MPa for up to three steps increased the efficiency of enzymatic hydrolysis on treated samples by increasing the concentration of N-acetyl-glucosamine. A reduction in chitin content was observed after 48 h of incubation in all HPH-treated samples inoculated with the yeasts, regardless of the strain applied. Specifically, the not-treated samples inoculated with *Y. lipolytica* RO25, PO11, and *D. hansenii* SP6L12 showed a chitin content of 12.19, 11.17, and 9.47 mg chitin /100 mg d.w., respectively, while the HPH-treated samples showed concentrations equal to 9.97, 9.58, and 9.15 mg chitin/100 mg d.w., respectively. These reductions, although following a trend, were not significant. This is partially not in agreement with Patrignani et al. [5] who demonstrated the high ability of *Y. lipolytica* and *D. hansenii* strains to degrade chitin after 72 h of incubation in a mixture of cricket powder and water. The main differences in that study compared to the current study were the method applied, that in our work may have generated chitin content data with higher variability, and the incubation time, which may have reduced the chitinolytic activity of the strains since only 48 h were applied instead of 72 h. Considering the data according to the strain, samples containing RO25 were the only ones in which the chitin content decreased significantly between the untreated and HPH-treated cricket solution. Even in Patrignani et al. [5], this strain promoted a higher reduction in chitin. 

### 3.3. Protein Characterization

The proteins were separated by SDS-PAGE under reducing conditions. The electrophoretic profile of the samples, shown in Figure 3, highlights some differences brought by both the biotechnological process and the HPH treatment. The greatest difference in the protein profile was observed between samples not inoculated and inoculated with the selected microorganisms. In particular, the most evident differences were observed in untreated samples after 24 and 48 h of incubation with *D. hansenii* SP6L12 (Db-NT), which had fewer evident bands at low molecular weights. The high proteolytic activity of *Y. lipolytica* and *D. hansenii* is widely reported in the literature [6,36,37]. The greatest decrease in the highest molecular weight bands was evident in HPH-treated samples inoculated and incubated for 48 h. NoH- samples at 0 h showed the impact of the HPH treatment. In fact, the protein pattern of the untreated sample was characterized by lower intensity bands having molecular weights between 45 and 31 kDa; on the contrary, bands between 31 and 21.5 kDa were more evident than in the HPH samples. When compared with the untreated samples, HPH-treated and inoculated samples presented a reduction in bands with molecular weights between 31 and 21.5 kDa in favor of bands with lower molecular weights. The literature is in line with the results obtained. In fact, the effect of HPH on protein hydrolysis and small-peptide production has been reported by several authors in different food matrices. Carullo et al. [38] reported that HPH caused a significant change in the conformation of milk proteins and affected the degree of enzyme hydrolysis, while Dong et al. [39] observed an increase in the degree of hydrolysis of the treated peanut protein isolate and an improvement in the production of low-molecular-weight peptides. It has been shown that cricket protein hydrolysates are a natural source of bioactive peptides and show less cross-reactivity towards serum IgE in shrimp allergy sufferers [40]. Moreover, the degree of hydrolysis of cricket proteins influences their sensory properties both directly by increasing bitterness and umami and indirectly by developing aromas following Maillard reactions [41]. In this frame, the combined use of HPH treatments and biotechnological processes could increase the nutritional value and the consumer acceptance of cricket flour when used as a food supplement to increase the protein content.

### 3.4. Thiolic Accessibility Characterization

To describe the impact of the HPH treatments on protein structure, the characterization of the thiol group accessibility was performed on NoH-NT and NoH-HPH samples. The data reported in Figure 4 show the slow kinetics of thiol titration. In fact, the amount of titrated thiols increased from 20 min to 1 h. Among the different samples analyzed under non-dissociating conditions (Figure 4), treatment with high pressure led to a decrease in accessible thiols, suggesting a rearrangement of the protein structure due to HPH. Rearrangements of protein structure, in turn, influence protein-related technological properties, such as solubility, water-binding capacity, foaming, and gelling properties [42] as well as accessibility to proteases leading to different (extent and nature) protein profiles [43]. Different studies have proved that milk, subjected to HPH pre-treatment at 100 MPa, had a higher yield in cheesemaking, identifying an increase in the water-binding capacity of proteins and in the different exposure of sulphydrilic compounds, the key factors in the highest yields [44]. In the presence of SDS, the quantity of titratable thiol groups increased, confirming the strong dissociating and solubilizing properties of this detergent on this kind of protein. The presence of SDS attenuated the differences between the two samples, suggesting that the anionic surfactant allows almost the total titration of the thiol groups contained in the samples. These results are in agreement with those reported by Santiago et al. [42], which highlighted that for the black cricket, the (thermal) aggregation of proteins is driven mainly by hydrophobic interactions, while the formation of intramolecular disulfide bridges is negligible [42].

### 3.5. Antioxidant Activity

The antioxidant activity was assessed using two analytical methodologies (ABTS and DPPH). HPH-treated (-HPH) and not-treated (-NT) cricket powder samples after 24 and 48 h of incubation with the selected strains (RO25-, PO11-, and Db-) were compared with the not-hydrolyzed cricket samples immediately after production and during a subsequent 48 h of incubation (NoH-NT). The HPH-treated sample immediately after production (NoH-HPH 0 h) was also used as a reference. In contrast to the DPPH method, which showed non-significant differences between the samples tested, the ABTS method demonstrated that the antioxidant capacity (TAC) of the samples analyzed was time-dependent. Although the ABTS and DPPH assays were both single-electron transfer-based assays and determined the anti-radical capacity of the molecules in the sample, differences were found between the two methods. These discrepancies may be due to the antioxidants’ solubility in the reaction media (water and hydro-methanolic solution in the ABTS and DPPH assays, respectively) and electron transfer kinetic issues [45]. Figure 5 shows that, regardless of the treatment and strain used, the antioxidant activity of the samples increased over time when compared to the samples immediately after preparation, with few exceptions. The NoH-NT and NoH-HPH samples immediately after preparation (0 h) proved that HPH treatment did not affect the TAC of the starting matrix. On the contrary, the HPH treatment, when combined with the biotechnological process obtained by selected yeasts, increased the antioxidant activity of the final hydrolyzed matrix. In particular, after 48 h, Db-HPH and PO11-HPH showed a higher increase in TAC than the not-treated samples Db-NT and PO11-NT. This is in line with previous studies showing that HPH has great potential to be used to improve the extractability and maintain the stability of bioactive compounds such as carotenoids, vitamin C, and polyphenols from a variety of substrates due to the alteration of the physical structures of food [46,47].

In contrast, RO25-NT, after 48 h, showed TAC values similar to RO25-HPH, PO11-HPH, and Db-HPH. The strong proteolytic activity of *Y. lipolytica* RO25 was documented by Patrignani et al. [5] and, regardless of HPH treatment, the RO25 strain can favor the formation of peptides endowed of antioxidant activity [48]. Although there is growing interest in the hunt for antioxidant peptides in edible insects, the mechanism by which the peptides exert their antioxidant activity is not fully known. Notably, various amino acids, such as Hys, Pro, Tyr, and Trp, have antioxidant properties, and these amino acids are typically found in peptide sequences [49]. Liu et al. also showed that low-molecular-weight peptides have more amino acids exposed to interact with free radicals, which boosts their antioxidant activity [50].

An unexpected result was the not-inoculated sample after 48 h of incubation (NoH-NT), which showed a TAC value of 0.67 mM TE. Despite data collected after 24 h, which confirmed the biotechnological process efficiency on the antioxidant compound production, after 48 h of incubation, the TAC value was statistically similar to those of samples inoculated with the yeasts and treated with HPH. It is important to mention that the cricket powder used in this study was characterized by a native microflora derived from the raw material. The absence of selected microorganisms in the mixture prevented the creation of selective conditions and favored the spontaneous and uncontrolled fermentation of the cricket powder and water mixture. Patrignani et al. [5] showed the presence of yeasts and mesophilic bacteria, including lactic acid bacteria in a dough composed of cricket powder and water. The metabolic activity of native lactic acid bacteria could promote the increase in antioxidant capacity of the matrix. The obtained results suggested that the combination of *Y. lipolytica* RO25 and HPH treatment favored the increase in the antioxidant capacity of the cricket powder hydrolysates. Therefore, we cannot exclude that the activity observed may have been impacted by native microorganisms. The ability of HPH treatment to increase the susceptibility to proteolysis of different food matrices has been reported in the literature [51,52,53]. In fact, HPH treatment not only promotes the deconstruction of the raw material but also provides nutrients, including proteins, making them more accessible to extracellular hydrolases and promoting the interaction between substrate and yeast enzymes. As a result, sample proteins were hydrolyzed more effectively by the proteolytic metabolism of selected microorganisms.

## 4. Conclusions

To the best of our knowledge, this work represents the first study in which HPH and a biotechnological process are combined to improve the characteristics of cricket powder. HPH treatment not only reduced the natural occurring yeasts present in the matrix but promoted the growth of added strains of *Y. lipolytica* and *D. hansenii*. The sample treated with HPH and inoculated with *Y. lipolytica* RO25 was the only one that significantly reduced the chitin content with respect to the same sample not treated, although a similar trend was observed also with the other yeasts. A longer incubation time might further reduce the chitin content. The antioxidant capacity of the initial matrix was not affected by HPH treatment; rather, it was dependent on the microbial strain used. The use of the sole strain *Y. lipolytica* RO25 slightly increased the antioxidant capacity of the mixtures, while the combination of HPH treatment and the biotechnological process imparted the greatest antioxidant capacity to the hydrolysates after 48 h of incubation. Specifically, the highest antioxidant activity was recorded in samples treated with HPH and fermented with *Y. lipolytica* PO11 and *D. hansenii* SP6L12. The combination of the two approaches also affected the protein pattern of cricket powder by decreasing the intensity of high-molecular-weight bands while increasing those of lower molecular weights. In addition, the HPH treatment caused a decrease in accessible thiols resulting in a reorganization of protein structure that could modify technological properties, such as solubility or water-binding capacity. Further studies are required to clarify how the modification induced by the yeasts applied in this study combined with HPH could impact the final technological properties of the product. This latter aspect will be fundamental to defining the proper processes for transformation of these materials in food. 

## Figures and Tables

**Figure 1 foods-13-00449-f001:**
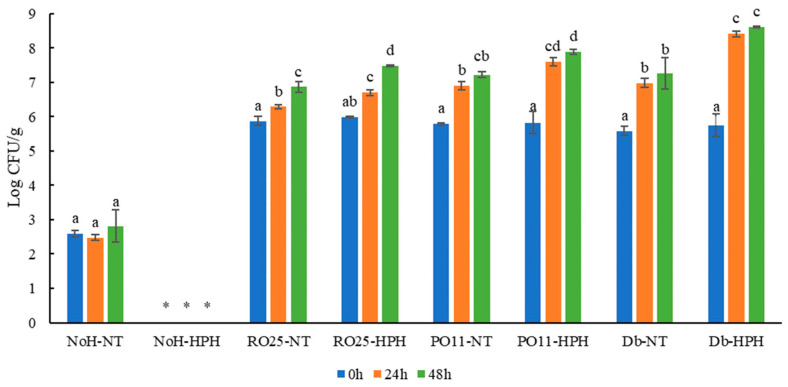
Cell loads (Log CFU/g) of cricket powder not-inoculated (NoH) or inoculated with *Y. lipolytica* RO25 (RO25), *Y. lipolytica* PO11 (PO11), and *D. hansenii* SP6L12 (Db) and treated (-HPH) and untreated (-NT) with high-pressure homogenization immediately after incubation (0 h) and after 24 and 48 h of incubation at 25 °C with agitation. For each microbial strain tested under different conditions, different letters indicate significantly different values. * Under the detection limit.

**Figure 2 foods-13-00449-f002:**
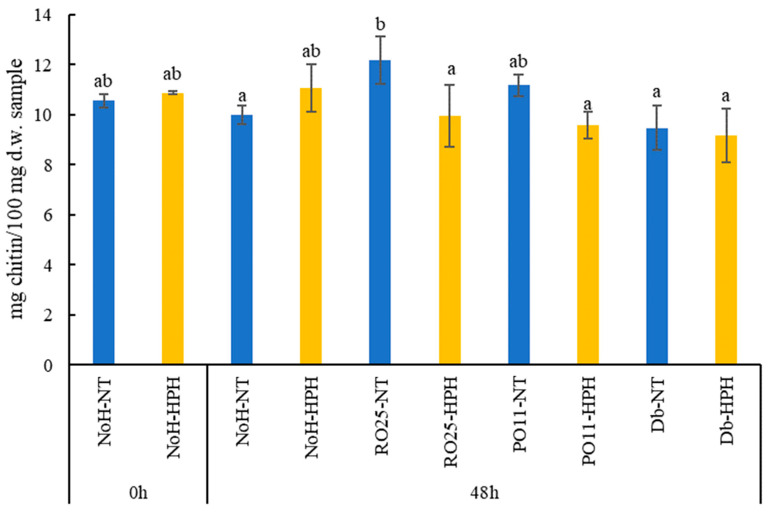
Chitin content (mg/100 mg dry weight (d.w.)) of cricket powder not-inoculated (NoH-) and inoculated with *Y. lipolytica* RO25 (RO25-), PO11 (PO11-), and *D. hansenii* SP6L12 (Db-) treated (-HPH) and not treated (-NT) with high-pressure homogenization immediately after incubation (0 h) and after 48 h of incubation at 25 °C with agitation. Different letters indicate significantly different values.

**Figure 3 foods-13-00449-f003:**
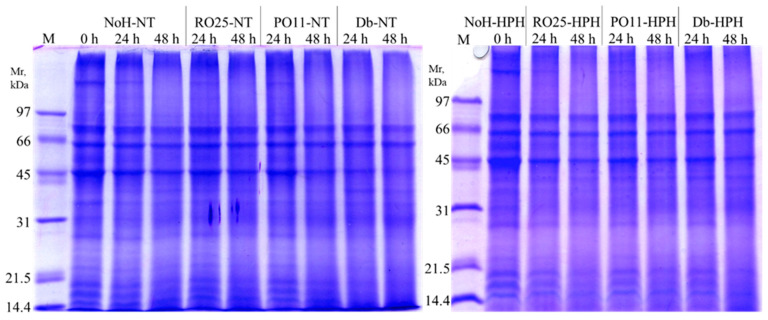
SDS-PAGE electrophoretic profile of cricket powder not-inoculated (NoH) or inoculated with *Y. lipolytica* RO25 (RO25), *Y. lipolytica* PO11 (PO11), and *D. hansenii* SP6L12 (Db) treated (-HPH) and untreated (-NT) with high-pressure homogenization immediately after incubation (0 h) and after 24 and 48 h of incubation at 25 °C with agitation.

**Figure 4 foods-13-00449-f004:**
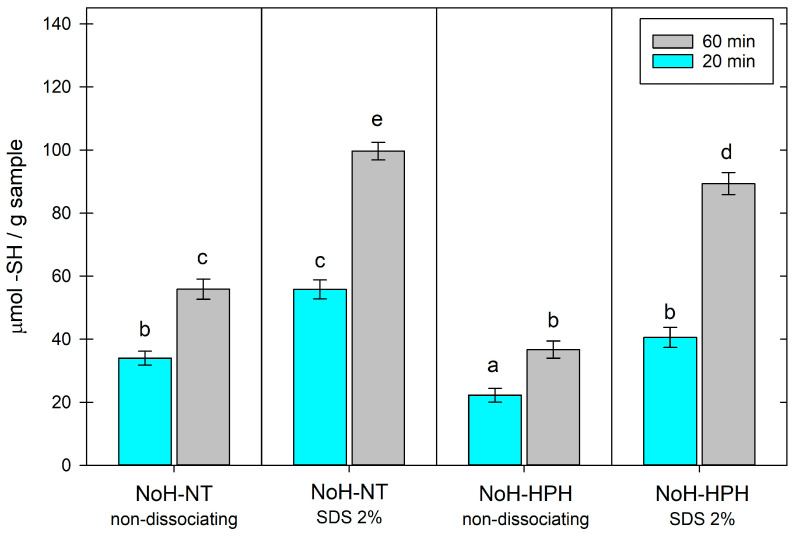
Accessibility of thiol groups in non-dissociating conditions and in the presence of SDS 2% after 20 min (cyan) and 60 min (gray) of incubation with DTNB. NoH-NT: not-inoculated and not-treated sample; NoH-HPH: not-inoculated and HPH-treated sample. Different letters indicate significantly different values.

**Figure 5 foods-13-00449-f005:**
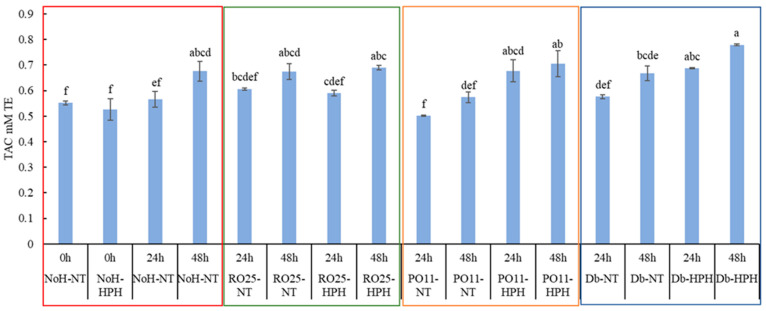
Antioxidant capacity (TAC) expressed in mM Trolox equivalents (TE), determined by ABTS of cricket powder not-hydrolyzed (NoH-) or inoculated with *Y. lipolytica* RO25 (RO25-), *Y. lipolytica* PO11 (PO11-), and *D. hansenii* SP6L12 (Db-) treated (-HPH) and untreated (-NT) with high-pressure homogenization immediately after incubation (0 h) and after 24 and 48 h of incubation at 25 °C with agitation. Different letters indicate significantly different values.

## Data Availability

All the data presented in this study are available in the manuscript.

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
