# Peer review of "Effect of Combined High-Pressure Homogenization and Biotechnological Processes on Chitin, Protein, and Antioxidant Activity of Cricket Powder-Based Ingredients"

_foods, 2024, doi:10.3390/foods13030449_

Round 1
Reviewer 1 Report
Comments and Suggestions for Authors
In the study entitled as “Effect of combined High-Pressure Homogenization and biotechnological processes on chitin, protein and antioxidant activity of cricket powder-based ingredients” has focused on the use of high-pressure homogenization on the cricket powder-water matrix. After the HPH application the matrices were inoculated by 2 strains of Y. lypolitica and one strain of D. hansenii. The effect of this combined process on the microbiological quality, structural organization of proteins, chitin content and antioxidant activity were followed.
Main weakness of the study is the lack of parameters of HPH other than 180 MPa.
In the title, I recommend the use of “fermentation” instead of “biotechnological process”
No numerical result was given in the abstract.
What is the motivation of applying HPH? The sentences given in the introduction about this point are not satisfactory. Please revise especially regarding its effect on microstructure and fermentation.
Discussion in 3.1. should be improved. No comments given on the effect of HPH on microstructure.
Discussion in 3.2. should be improved. More detailed comments are required on the effect of fermentation on chitin degradation.
Please find my extra comments on the comment section of the uploaded pdf.

Author Response
We want to thank reveiwer 1 for the comments and suggestions that we considered, when possibile. Please find attached the point by point answers to reviewer1

Reviewer 2 Report
Comments and Suggestions for Authors
This paper entitled “Effect of combined High-Pressure Homogenization and biotechnological processes on chitin, protein and antioxidant activity of cricket powder-based ingredients”. The authors present some interesting and useful information for readers. Overall the objectives and rationale are clear and well structured. However, they must clarify some unclear points and some additional information is needed for clarification.
Main Comments:
1. L. 21-31: Some data and important results should be added in abstract.
2. L. 89: Since HPH is important technique in this experiment, please describe carefully how to treated them, chosen the pressure, as well as detail method.
3. L. 90, 91, 131, 198, 226, 319, 330: Please uniform the reference mark in throughout text.
4. L. 160: please check the unit of centrifugation.
5. L. 206, 343: Please check the correctness of the statistical data and labeling in Figures 1 & 5.
6. L. 295: Please consider modifying the horizontal axis of Figure 4 to make it easier for readers to understand.
7. All results should be better discussed respect data present in literature and innovative character should be better marked.

Author Response
We want to thank reveiwer 2 for the comments and suggestions that we considered, when possibile. Please find attached the point by point answers to reviewer2

Reviewer 3 Report
Comments and Suggestions for Authors
The proposed article „Effect of combined High-Pressure Homogenization and biotechnological processes on chitin, protein and antioxidant activity of cricket powder-based ingredients” prepared by Samantha Rossi 1, Davide Gottardi 1, 2, *, Alberto Barbiroli 3, Mattia Di Nunzio 3, Lorenzo Siroli 1, 2, Giacomo Braschi 5 1, Oliver Schluter 1, 4, Francesca Patrignani 1, 2, Rosalba Lanciotti 1, 2, refers to a topic concerning the planet’s increasing population leading to researching of more options for food ingredients. Insects, being rich in protein, fat, mineral, vitamin, and energy content, would be useful alternative to address the expected animal protein-based product demand. The authors suggest the combination of high-pressure homogenization of insect materials and biotechnological process, carried out by different yeast strains, in order to evaluate the effect on production of innovative ingredients with high functional and technological properties.
The manuscript’s topic is relevant to the current needs of the society in regard to the world’s problem with food deficiency and the necessity to utilize more substances in food for humans as well as for animals.
Here are my comments:
The list of references is quite short; 47% of them are from the last 10 years. Also, the self-citations are almost 30%.
In my opinion, the article would benefit from more recent studies mentioned, and more detailed discussion in the section Results and discussion for each subsection – in most cases, stating that other studies confirm the results is not enough for discussion.
The paper work needs the following corrections:
- Lines 157, 172, 190, 224, 367 in the PDF file - my suggestions are colored in green with yellow notes.
- In the references list, all the references should be formatted according to the guidelines of the journal – the journal names should be abbreviated and in Italic font.
- In my opinion, the English language is quite good and needs just a few corrections.
As a reviewer, I state that I do not have conflicts of interest with the authors of this manuscript.

In my opinion, the English language is quite good and needs just a few corrections.
Author Response
We want to thank reveiwer 3 for the comments and suggestions that we considered, when possibile. Please find attached the point by point answers to reviewer3

Round 2
Reviewer 1 Report
Comments and Suggestions for Authors
I have gone through each response of authors. I have also checked the responses embedded in the manuscript. As the Reviewer 1, I want to inform you that the Authors have successfully responded or revised the manuscript according to all my previous comments.
I think that as a result of this review process, the manuscript was developed enough.
Author Response
We want to thank reviewer 1 for the comments